# Ligand-Independent Vitamin D Receptor Actions Essential for Keratinocyte Homeostasis in the Skin

**DOI:** 10.3390/ijms26010422

**Published:** 2025-01-06

**Authors:** Satoko Kise, Shinichi Morita, Toshiyuki Sakaki, Hiroyuki Kimura, Seigo Kinuya, Kaori Yasuda

**Affiliations:** 1Department of Nuclear Medicine, Kanazawa University Hospital, Kanazawa University, 13-1 Takara-machi, Kanazawa 920-8641, Ishikawa, Japan; kinuya@med.kanazawa-u.ac.jp; 2Division of Probe Chemistry for Disease Analysis, Research Center for Experimental Modeling of Human Disease, Kanazawa University, 13-1 Takara-machi, Kanazawa 920-8460, Ishikawa, Japan; hkimura@staff.kanazawa-u.ac.jp; 3Department of Pharmaceutical Engineering, Faculty of Engineering, Toyama Prefectural University, 5180 Kurokawa, Imizu 939-0398, Toyama, Japan; kyasuda@pu-toyama.ac.jp; 4Division of Evolutionary Developmental Biology, National Institute for Basic Biology, 38 Nishigonaka, Myodaiji, Okazaki 444-8585, Aichi, Japan; shinichi@nibb.ac.jp; 5Department of Basic Biology, School of Life Science, The Graduate University for Advanced Studies, SOKENDAI, Hayama, Miura 240-0115, Kanagawa, Japan

**Keywords:** vitamin D receptor, keratinocyte homeostasis, ligand-independent VDR action, hyperkeratosis, alopecia

## Abstract

Recently, we demonstrated that the alopecia observed in vitamin D receptor gene-deficient (*Vdr*-KO) rats is not seen in rats with a mutant VDR(R270L/H301Q), which lacks ligand-binding ability, suggesting that the ligand-independent action of VDR plays a crucial role in maintaining the hair cycle. Since *Vdr*-KO rats also showed abnormalities in the skin, the relationship between alopecia and skin abnormalities was examined. To clarify the mechanism of actions of vitamin D and VDR in the skin, protein composition, and gene expression patterns in the skin were compared among *Vdr*-KO, *Vdr*-R270L/H301Q, and wild-type (WT) rats. While *Vdr*-R270L/H301Q rats exhibited normal skin formation similar to WT rats, *Vdr*-KO rats showed remarkable hyperkeratosis and trans-epidermal water loss in the skin. RNA sequencing and proteomic analysis revealed that the gene and protein expression patterns in *Vdr*-KO rats significantly differed from those in WT and *Vdr*-R270L/H301Q rats, with a marked decrease in the expression of factors involved in *Shh*, *Wnt*, and *Bmp* signaling pathways, a dramatic reduction in the expression of hair keratins, and a substantial increase in the expression of epidermal keratins. This study clearly demonstrated that non-liganded VDR is significantly involved in the differentiation, proliferation, and cell death of keratinocytes in hair follicles and the epidermis.

## 1. Introduction

Vitamin D is a multifunctional hormone and is usually obtained from foods like fish or synthesized in the skin when 7-dehydrocholesterol is cleaved by UV light, followed by isomerization in the body, to produce vitamin D_3_ (VD3). VD3 is initially hydroxylated in the liver by P450 enzymes such as CYP2R1 and CYP27A1 to form 25-hydroxyvitamin D3 (25D3), which is referred to as pro-vitamin D_3_. Then, 25D3 is converted to 1α,25-dihydroxyvitamin D_3_ (1,25D3), the active form of vitamin D3, by CYP27B1 in the kidneys. 1,25D3 binds to the vitamin D receptor (VDR) present in the cytoplasm, inducing VDR’s translocation to the nucleus. Subsequently, VDR forms a heterodimer with the retinoid X receptor (RXR), binds to vitamin D response elements (VDREs), and regulates the expression of genes involved in bone formation and calcium homeostasis (such as calcium channels, *osteocalcin*, and *osteopontin*). This mechanism is widely known as the classical genomic action of vitamin D [1] (Figure 1).

On the other hand, recent studies have highlighted the importance of non-genomic actions in vitamin D (Figure 1) [2,3,4] and ligand-independent VDR functions [5,6]. Rats with *Cyp27b1* gene deficiency (*Cyp27b1*-KO) exhibit much more significant decreases in serum calcium levels and growth inhibition compared to rats with vitamin D receptor gene deficiencies (*Vdr-KO*) [7]. This is thought to be based on the loss of non-genomic actions of 1,25D3 that are not mediated by VDR. Additionally, type II rickets caused by mutations in the *Vdr* gene often lead to alopecia. Mutations in the DNA-binding domain or RXR interaction site of VDR cause alopecia, whereas mutations in the ligand-binding domain do not. This suggests that ligand-dependent VDR action, i.e., the canonical genomic action of vitamin D, is not essential for the hair cycle.

To evaluate how vitamin D acts in various organs and cell types, we generated genetically modified rats (GM rats) with altered vitamin D-related genes. *Cyp27b1*-KO rats cannot produce 1,25D3, while *Vdr-KO* rats lack most of the ligand-binding domain of VDR [7,8]. Furthermore, we produced rats with mutant VDRs. VDR’s Arg270 forms a hydrogen bond with the 1α-hydroxyl group of 1,25D3, and His301 forms a hydrogen bond with the 25-hydroxyl group. Therefore, the VDR-R270L mutant, where Arg270 is replaced with Leu, has an affinity for 1,25D3 reduced to approximately 0.1% that of WT, and the VDR-H301Q mutant, where His301 is replaced with Gln, has an affinity for both 1,25D3 and 25D3 reduced to several tens of that of WT. Humans with corresponding VDR-R274L or VDR-H305Q mutations exhibit type II rickets without alopecia.

We also created rats having the double mutant *Vdr*-R270L/H301Q, which has almost no ability to bind both 1,25D3 (less than 0.01% of that of VDR-WT) and 25D3 (less than 1% of VDR-WT). The *Vdr*-R270L/H301Q rats appear to be the best animal model for analyzing the function of ligand-independent VDR [6,8]. As shown in Figure 2, comparing WT rats with *Vdr*-R270L/H301Q rats highlights the effects of ligand-dependent VDR. Additionally, comparing *Vdr*-R270L/H301Q rats with *Vdr-KO* rats reveals the effects of ligand-independent VDR. Since only *Vdr-KO* rats exhibit alopecia with age, this indicates that ligand-independent VDR action plays a crucial role in the hair cycle. On the other hand, bone formation abnormalities are observed in both *Vdr*-R270L/H301Q and *Vdr-KO* rats, suggesting that ligand-dependent VDR action is important for bone formation.

Vitamin D contributes to hair growth during the anagen phase of the hair cycle by activating the *Wnt* signaling pathway and is involved in maintaining skin homeostasis and wound healing [9]. Keratinocyte-specific *Vdr* gene-deficient KO mice, generated by crossing mice expressing Cre recombinase under the control of the *Krt14* promoter with mice carrying loxP sites flanking the *Vdr* gene, exhibit alopecia and impaired wound healing [9]. From these results, it was hypothesized that VDR is necessary for wound healing, though more so for keratinocyte migration rather than proliferation [10]. On the other hand, Tian et al. [11] found that the administration of 1,25D3 was effective for wound repair, suggesting a ligand-dependent VDR action. Calcium and 1,25D3 were thought to be necessary for keratinocyte differentiation and cornified envelope formation [12]. While the involvement of CaSR, PLC, PKC, and AP-1 factors has been suggested, the involvement of VDR remains unclear. Thus, how VDR is involved in keratinocyte proliferation and migration is not yet fully understood. Recently, Joko et al. [13] suggested that VDR plays a crucial role in inducing keratinocyte apoptosis during the catagen phase of the hair cycle. They showed the presence of “surviving epithelial strands” in the hair follicles of *Vdr*-KO mice, which results in dermal cyst accumulation. In this study, to clarify whether the skin abnormalities in *Vdr*-KO rats that follow alopecia are related to the dermal cyst formation or arise through completely different mechanisms, we attempted to elucidate the function of VDR in maintaining skin homeostasis by evaluating differences in skin phenotypes, functions, gene expression, and protein expression among WT, *Vdr*-R270L/H301Q (hereafter simply referred to as KI), and *Vdr*-KO (KO) rats.

## 2. Results

### 2.1. Comparison of Skin Morphology from Young Stage to Older Stage Among WT, KI, and KO Rats with HE Staining

The skin morphology of WT, KI, and KO rats was analyzed across different developmental stages. At the young stage (4 weeks), no discernible differences were observed among the three groups. However, by the middle stage (7 weeks) and older stage (22 weeks), significant skin abnormalities, including cyst formation and hyperkeratosis, were evident exclusively in KO rats (Figure 3). These findings indicate that ligand-independent VDR functions play a crucial role not only in maintaining the hair cycle but also in preserving skin structure and function.

### 2.2. VDR Localization in the Rat Skin via Immunofluorescence Analysis

VDR is known to be expressed in keratinocytes and in the lower proximal region of hair follicles. To assess the localization of KI in the skin of KI rats, immunofluorescence analysis was conducted on skin samples from 7-week-old rats. Similarly to WT and KI rats, VDR expression was exhibited in keratinocytes and in the lower proximal cup of the hair follicle. In contrast, no VDR signal was detected in the skin of KO rats (Figure 4).

### 2.3. SDS-PAGE Analysis of Rat Skin Proteins with CBB Staining and Western Blot Analysis Using Anti-Krt14 and Anti-Pan-Keratin Antibodies

To evaluate the overall protein composition in the skin of each rat, SDS-PAGE followed by CBB staining was performed on skin samples from 4-, 15-, and 30-week-old rats. At the young stage (4 weeks), no significant differences were observed among the three groups (Figure 5). However, by the middle stage (15 weeks) and older stage (30 weeks), the protein pattern in KO rats showed marked differences compared to both WT and KI rats, with a notable disparity around the 55 kDa region. Given the hyperkeratosis observed in VDR-KO rats through HE staining, we conducted Western blotting using the antibody against the keratin marker Krt14 and anti-pan-keratin antibody (PCK26), which detects rat keratin 1 (66 kDa), keratin 5 (58 kDa), and keratin 8 (52 kDa). Consistent with expectations, KO rats exhibited a significant overexpression of these keratin markers (Appendix A).

### 2.4. Trans Epidermal Water Loss (TEWL) in the Rat Skin

TEWL refers to the passive evaporation of water from the skin into the external environment. It serves as a valuable indicator for assessing skin health and hydration levels and is commonly used to evaluate skin disease models [14]. Typically, low TEWL values reflect a healthy and intact skin barrier, whereas elevated TEWL values suggest a compromised barrier function. Across all stages examined (5, 10, and 30 weeks), *Vdr*-KO rats consistently exhibited higher TEWL values compared to the other groups, indicating impaired skin barrier integrity (Figure 6). 

### 2.5. Gene Expression Analysis by Bulk RNA-Seq

The symptoms of hair loss, skin thickening, and cyst formation were observed exclusively in KO rats and not in WT or KI rats (Figure 3 and Appendix A). Additionally, transepidermal water loss (TEWL) was significantly higher only in KO rats, suggesting a decline in skin barrier function (Figure 6). Furthermore, in 30-week-old KO rats, a remarkable expression of the epidermal keratin, Krt14, was observed (Figure 5). These results suggest that the gene expression profile in KO rats differs significantly compared to WT and KI rats. Thus, we analyzed nine RNA-seq libraries created from the skin tissues of three groups (WT, KO, and KI groups, with three rats in each group). We quantified the gene expression levels and performed hierarchical clustering analysis on the nine samples. As a result, samples within each group clustered together, with each group exhibiting a distinct gene expression pattern (Figure 7A).

We then performed two different types of comparisons to identify transcripts involved in keratinocyte homeostasis under the regulation of VDR (FDR < 0.05). First, we compared the WT and KO groups to comprehensively identify genes regulated by VDR in both ligand-dependent and ligand-independent manners. This analysis identified 10,015 DEGs (Figure 7B). Subsequently, to identify genes regulated by VDR in a ligand-dependent manner, we compared the WT and KI groups (Figure 2). This comparison identified 880 DEGs. Many of the 880 DEGs overlapped with those obtained in the WT vs. KO comparison, indicating the accuracy of this analysis (Figure 7B). In addition, the KI vs. KO comparison, which highlights ligand-independent VDR action (Figure 2), revealed 4886 DEGs. It is also noteworthy that the number of DEGs identified in the WT vs. KI comparison was significantly smaller than that in the KI vs. KO comparison. These findings suggest that most genes regulated by VDR in the skin are likely controlled by non-liganded VDR.

Figure 8 compares the expression of the *Shh* pathway (*Shh*), *Wnt* signaling pathway (*Cttnb1* [*β-catenin*], *Lef1*, *Wnt10b*, *Wise*), and *Bmp* pathway (*Bmp4*, *Noggin*). The results indicate a pronounced decrease in gene expression in KO rats compared to WT and KI rats, with the exception of *Wise*, known as an inhibitor of both the *Wnt* and *Bmp* signaling pathways [15]. This suggests that the *Wnt* signaling pathway is significantly suppressed in KO rats relative to WT and KI rats. The *Bmp* pathway inhibitor *Noggin* is also reduced in KO rats. However, considering the collective influence of *Bmp4*, *Noggin*, and *Wise*, it can be deduced that the progression of the *Bmp* pathway is likewise hindered in KO rats. These pathways are essential for sustaining the hair cycle [16], and the observed inability of KO rats to maintain this cycle further supports this conclusion. As depicted in Figure 8, the gene expression levels in KI rats closely resembled those in WT rats, consistent with the normal hair cycle maintenance observed in KI rats (Appendix A).

Figure 9 highlights keratins that were significantly increased only in KO (Appendix A, Appendix A). These keratins are all known as epidermal keratins and are located in the epidermis. The epidermis consists of the basal, spinous, granular, and cornified layers. *Krt5* and *14* are mainly expressed in the basal layer, while *Krt1* and *10* are expressed in the spinous, granular, and cornified layers. After hair loss, the disappearance of hair follicle keratinocytes and differentiation into epidermal keratinocytes, as well as cyst formation and skin thickening due to proliferation, led to the formation of significantly different skin in 30-week-old KO rats compared to WT and KI.

Figure 10 lists the genes essential for keratinization and the formation of the cornified envelope that showed a marked increase in expression in KO compared to WT and KI (Appendix A). The protein encoded by *Sprr1a* is involved in epithelial protection and repair and is upregulated in response to inflammation and stress. Transglutaminases encoded by *Tgm1* and *Tgm3* are enzymes that crosslink proteins, strengthening the structure of the cornified layer and providing durability. TMEM79 (transmembrane protein 79) is a transmembrane protein associated with skin barrier function, particularly in skin protection, allergic reactions, and eczema. These genes are all related to enhancing the skin’s barrier function. In KO rats, barrier function appears compromised, and their expression may have been induced to compensate.

Figure 11 lists the keratins that were significantly reduced only in KO rats. The hair follicle comprises the hair shaft, inner root sheath, and outer root sheath. Among the keratins in Figure 9, *Krt31*, *32*, *34–36*, *39*, *40*, and *81–85* are present in the hair shaft and are collectively known as hair keratins. *Krt25–28*, *71*, and *72* are keratins located in the inner root sheath [17]. The fact that these were significantly reduced in KO rats, and that hair loss and abnormal follicles were observed in KO rats, is in good agreement with the findings (Appendix A).

Figure 12 presents inflammation-related genes (*IL1β*, *S100a8*, *S100a9*) and a cell proliferation gene (*Mki67*) that showed increased expression only in KO. The skin of 30-week-old KO rats exhibits conditions reminiscent of symptoms seen in atopic dermatitis or psoriasis (Appendix A), prompting an investigation of inflammation-related gene expression. *IL-1β* (*Interleukin-1 β*) is a potent pro-inflammatory cytokine that plays a central role in immune response and inflammation. This protein is primarily produced by immune cells such as macrophages, neutrophils, and monocytes and is released in response to stress signals from infection, trauma, or tissue damage. S100a proteins are calcium-binding proteins involved in cellular proliferation and differentiation. They are notably expressed in immune cells and are upregulated during inflammation to modulate immune response to infection and inflammation. *Mki67* is an excellent marker for indicating cell proliferation, as it is only expressed during cell division. In 30-week-old KO rats, a wavy skin structure is observed (Appendix A), suggesting abnormal proliferation of keratinocytes. From these results, it is inferred that the skin of KO rats likely exhibits chronic inflammation and increased proliferation of epidermal keratinocytes.

## 3. Discussion

Previous reports have shown that vitamin D has anti-inflammatory, anti-aging, and wound-healing properties in the skin [9,18,19]. However, it is unclear whether these effects are mediated through VDR or independent of VDR. In keratinocytes in the skin, vitamin D_3_ is synthesized from 7-dehydrocholesterol by sunlight exposure, and 25-hydroxylation by CYP2R1 and CYP27A1 in keratinocytes, followed by 1α-hydroxylation by CYP27B1, produces active vitamin D_3_ [20,21]. Furthermore, it is believed that both calcium and 1,25D3 synthesized in keratinocytes are necessary for keratinocyte differentiation and cornified envelope formation. Although CaSR, PLC, PKC, and AP-1 factor involvement have been suggested in this mechanism, the role of VDR remains uncertain. Meanwhile, Tian et al. [11] found that the administration of 1,25D3 was effective in wound repair and considered it an action of ligand-dependent VDR. On the other hand, based on the decreased expression of *β-catenin* signaling target genes such as *Padi1*, *Dix*, and *Tubb3* in *Vdr*-cKO mice, VDR is thought to form a complex with β-catenin and Lef1 to activate *β-catenin* signaling and promote keratinocyte proliferation [1]. This action of VDR is speculated to be ligand-independent. Furthermore, recent studies by Jokoh et al. [13] suggested that VDR plays an essential role in inducing keratinocyte apoptosis during the regression phase of the hair cycle. Thus, the mechanisms by which vitamin D and VDR affect keratinocyte metabolism, active vitamin D production, and keratinocyte proliferation, differentiation, and apoptosis are complex and remain unclear. We conducted this study to identify the most critical mechanisms among these actions.

One notable point of this study is the use of rats with VDR that do not bind ligands. VDR (R270L/H301Q) hardly binds to 1,25D3 and 25D3. We have confirmed that physiological concentrations do not cause nuclear translocation or induce *Cyp24a1* transcription. In the previous studies, mouse VDR (L304H) [22] and human VDR(L233S) [23] have been regarded as lacking ligand-binding ability so far. However, the affinity of mouse VDR(L304H) for 1,25D3 appears to be approximately 1/10 of the WT-VDR. Similarly, studies using transgenic *Vdr*-KO mice expressing human VDR(L233S) specifically in keratinocytes appear to retain some extent of ligand-dependent VDR actions. In contrast, our rats differ significantly from the reported VDR mutants in that the double mutation of the two amino acid residues (Arg270 forms a hydrogen bond with the 1α-hydroxy group, and His301 forms a hydrogen bond with the 25-hydroxy group) contributing most to 1,25D3 binding results in the almost complete loss (less than 0.01%) of 1,25D3-dependent VDR function. It is noted that all enzymes required for synthesizing active vitamin D are present in keratinocytes [21]; then the concentration of 1,25D3 might be higher in keratinocytes than in the blood or other organs. To precisely differentiate between ligand-dependent and ligand-independent actions of VDR in keratinocytes, it is essential to use a *Vdr* variant that completely lacks ligand-binding capability. *Vdr*-R270L/H301Q appears to be the most suitable candidate for this purpose. Comparing the skin of *Vdr* (R270L/H301Q) rats with *Vdr*-KO rats will highlight the ligand-independent actions of VDR in keratinocytes. To the best of our knowledge, the concentration of VDR ligands in the skin has not been previously reported. However, our RNA sequencing results demonstrated *Cyp24a1* expression in KO rats, while no expression was observed in both WT and KI rats. This suggests that the concentration of VDR ligands in WT rat skin is exceedingly low. In addition, repression of *Cyp24a1* gene transcription by non-liganded VDR likely occurred in both WT and KI rats, as previously reported [24].

In HE staining, cyst formation was observed only in the 30-week-old KO rats, while KI rats were similar to WT rats. Western blot analysis showed a significant reduction in Krt14 in 4-week-old KO rats compared to WT and KI rats (Figure 3), consistent with the findings of Xie et al.’s qPCR analysis, which indicated reduced keratinization in 3-week-old VDR-KO mice [25]. No apparent skin abnormalities and normal hair growth were seen in 4-week-old KO rats. However, at 15 weeks, the amount of Krt14 protein was significantly higher than in WT and KI rats, and it was markedly elevated at 30 weeks. The levels of Krt1 and Krt5 proteins were also significantly increased in 30-week-old KO rats (Appendix A). Consequently, RNA-seq was used to compare the gene expression among WT, KI, and KO rats. First, when examining the expression of genes essential for hair cycle regulations, their remarkable reduction was observed only in KO rats (Figure 8). Similarly, in 11-week-old *Vdr*-KO mice [16], reductions in *Shh*, *Lef1*, *Wnt10b*, *Bmp4*, and *Noggin* were reported. In our current findings, a significant reduction was also observed in *β-catenin* gene expression, along with notable decreases in the *Shh* pathway (*Shh*) and the *Wnt* signaling pathway (*Ctnnb1* (*β-catenin*), *Lef1*, *Wnt10b*). Conversely, *Wise*, an inhibitor of both the *Wnt* and *Bmp* signaling pathways, was increased in KO rats. Therefore, it can be inferred that the *Wnt* signaling pathway is markedly suppressed in KO rats. Additionally, *Bmp4* was significantly decreased in the KO rats, while its inhibitor *Noggin* was also reduced. Considering the increase in wise in KO rats, it is possible to assume that this pathway may also be downregulated in KO rats. No significant differences were observed between KI and WT rats in these pathways, indicating that ligand-independent VDR may play a crucial role in the expression of *Shh*, *Wnt*, and *Bmp* pathway-related genes.

The *Hr* mRNA level was higher in KO rats compared to WT and KI rats, which is consistent with the fact that *Hr* expression was higher in 11-week-old VDR-KO mice than in WT mice [16].

Comparison of keratin expression among WT, KI, and KO rats revealed a remarkable decrease in KO rats in the expression of 18 types of hair keratin and the keratins specifically expressed in the inner root sheath [17,26]. These results are consistent with the absence of hair follicles in KO rats as shown in Figure 3. In contrast, the expression of 12 types of epidermal keratins was dramatically upregulated in KO rats, suggesting the enhanced proliferation of keratinocytes forming the epidermis layers (basal spinous, granular, and cornified). In addition, a proliferation of Krt10-positive cyst-forming keratinocytes [13] may be a concern. These findings are consistent with Western blot analysis (Krt14 in Figure 4, Krt1 and Krt5 in Appendix A) and proteomic analysis (Krt24 and Krt80 in Appendix A). Teichert et al. [16] also reported a reduction in the expression of 5 hair keratins and an increase in 2 epidermal keratins in *Vdr*-KO mice. Additionally, KO rats showed a marked increase in gene expression related to keratinization and cornified envelope formation, essential for skin barrier function (Figure 6), consistent with proteomic analysis (Appendix A). Given the importance of the cornified envelope in skin barrier function, it is likely that KO rats exhibit enhanced skin barrier function. However, as shown in Figure 6, only KO rats exhibited a significant increase in trans-epidermal water loss (TEWL). Since an intact skin barrier would suppress water loss, the elevated TEWL suggests impaired skin barrier function in KO rats. The findings in Figure 12 also imply potential chronic inflammation in KO rat skin, with compromised barrier function possibly resulting from chronic inflammation. Therefore, the observed increase in keratinization and cornified envelope-related genes in KO rats may represent a compensatory mechanism for the decreased barrier function. Interestingly, even at 5 weeks, KO rats showed higher TEWL than WT and KI rats. However, HE staining revealed apparently normal skin with no signs of alopecia. Nonetheless, Western blot analysis of 4-week-old KO rats (Figure 5) suggested that keratinization may be insufficient, and the barrier function might be low. This finding appeared to be consistent with the lower keratinization of 3-week-old *Vdr*-KO mice than WT-mice [25]. Although the TEWL values were high at 5, 10, and 30 weeks, we hypothesize that the underlying reasons differ between 5-week-old KO rats and 10- and 30-week-old rats.

Based on our current results and previous reports,

Non-liganded VDR is necessary for the expression of genes in *Shh, Wnt*, or *Bmp* pathways essential for hair cycle regulations.The initial hair cycle proceeds through anagen and catagen phases even in the absence of VDR.The transition from catagen to telogen requires VDR-dependent apoptosis of the epithelial strand of the hair follicle [13].Epithelial strands of hair follicles that evade apoptosis form cysts, which continue to proliferate to result in the formation of wavy skin surfaces as cornified layers (Appendix A).

Notably, in KO rats, there is a marked reduction in the key signaling pathways (*Shh, Wnt*, and *Bmp*), all of which play essential roles in the maintenance and differentiation of epidermal stem cells, hair follicle formation, regeneration, and wound healing. Therefore, KO rats likely experience a disorderly state where these regulatory processes are significantly less functional (Appendix A). It is noteworthy that the non-liganded VDR regulates the transcription of these crucial genes.

It is known that hairless (Hr) forms a complex with non-liganded VDR and binds to VDRE, suppressing gene expression [27]. However, we hypothesize that non-liganded VDR might have a mechanism promoting transcription of *Shh*, *Ctnnb1*, *Lef1*, *Wnt10b*, and *Bmp4* genes. This is quite an essential function of VDR, and we aim to investigate this further. It is noted that compounds that promote *Shh, Wnt*, and *Bmp* signaling pathways have the potential to become therapeutic agents for alopecia and skin-related disorders derived from VDR deficiency.

## 4. Materials and Methods

### 4.1. Animals and Diets

Jcl:Wistar rats were obtained from CLEA Japan Inc. (Tokyo, Japan). Embryonic microinjection for genome editing was performed by KAC Co., Ltd. (Kyoto, Japan). All of the rats were kept at room temperature (22–26 °C) and in 50 to 55% humidity with a 12 h light/dark cycle. They were allowed food and water ad libitum and fed CE-2 (Oriental Yeast Co., Tokyo, Japan). Homozygotes of KO rats were maintained by mating homozygotes. Genotype was determined by electrophoresis of PCR products of the target site for KO rats. All of the experiments were conducted with male rats. All of the experimental protocols using animals were performed in accordance with the Guidelines for Animal Experiments at Toyama Prefectural University and were approved by the Animal Research and Ethics Committee of Toyama Prefectural University.

### 4.2. Immunofluorescence

Each rat skin sample was fixed with 4% PFA (FUJIFILM Wako Pure Chemical Corporation, Tokyo, Japan) for 15 h at 4 °C. The fixed samples were mounted with O.T.C. compound (Sakura Finetek Japan, Tokyo, Japan) in a container and frozen in liquid nitrogen. The frozen sections were prepared with a cryostat microtome (Leica, Tokyo, Japan) with a thickness of 20 µm for each sample and stuck it glass slides. Anti-VDR antibody (D2K6W) Rabbit mAb (Cell Signaling Technology, Danvers, MA, USA), and Alexa Fluor 488 goat anti-rabbit IgG (Invitrogen, Carlsbad, CA, USA) were used for immunofluorescence [28]. After staining nuclei with DAPI, samples were observed using phase contrast microscopy (Olympus, Tokyo, Japan).

### 4.3. HE Staining

Each rat skin sample was fixed with 4% PFA (FUJIFILM Wako Pure Chemical Corporation, Osaka, Japan) for 15 h at 4 °C. The fixed samples were mounted with O.T.C. compound (Sakura Finetek, Japan) in a container and frozen in liquid nitrogen. Frozen sections were obtained by cryostat microtome (Leica, Tokyo, Japan) with a thickness of 20–25 µm, and stuck to glass slides [7]. The 4% PFA solution was dripped into the sample glass, which was incubated at room temperature for 10 min, and washed with water for 10 min. Hematoxylin was dripped on the sample glass, which was incubated at room temperature for 15 min, and washed with water for 10 min. Eosin-alcohol was dripped on the sample glass, which was incubated at room temperature for 2 min, and washed with water for 2 min, 70% EtOH for 2 min, 80% EtOH for 2 min, 90% EtOH for 2 min, 100% EtOH for 2 min, and then washed with xylene twice. The resultant HE-stained samples were observed using phase contrast microscopy (Olympus, Tokyo, Japan).

### 4.4. Western Blot Analysis

After the back hair of rats was shaved with a hair clipper, the skin was cut out with scissors and then homogenized with a Minylis personal homogenizer (Bertin Technologies, Montigny-le-Bretonneux, France). The resultant tissue lysate containing 20 µg protein was applied to each lane of the gel and then subjected to SDS–PAGE on 4 to 20% linear gradient polyacryl amide/SDS gels. Molecular size marker (blue pre-stained protein standard, Broad Range P7718S, New England BioLabs. Inc., Ipswich, MA, USA) was loaded with 3 μL. After electrophoresis, gels were electrotransferred onto PVDF membranes. The membranes were incubated in TBS-T containing 5% skim milk and then incubated with the 1st antibody (Table 1). The membranes were washed three times with Tris-buffered saline containing 0.05% Tween 20 (TBS-T) and incubated with horseradish peroxidase-conjugated goat anti-rabbit IgG (Cell Signaling Technology, Danvers, MA, USA). The membranes were washed with Tris-buffered saline containing 0.05% Tween 20 (TBS-T) and then followed by the enhanced chemiluminescence immunodetection method (Amersham Pharmacia Biotech, Buckinghamshire, England).

### 4.5. Measurement of Moisture Transpiration (TEWL)

Rat’s back hair was shaved with a hair clipper, and TEWL was detected with VAPOSCAN AS-VT100 (Asch Japan Co., Ltd., Tokyo, Japan). The TEWL is calculated as the amount of water vapor flux per unit area of skin (usually expressed in g/m^2^/h). This is based on the following formula derived from Fick’s Law:TEWL = D × dC/dx

D is the diffusion coefficient of water vapor in air. dC/dx is the concentration gradient of water vapor measured by the sensors [14].

### 4.6. RNA-Seq Analysis

The back hairs of the 30-week-old male rats (WT, KO, and KI groups, with 3 rats in each group) were shaved with hair clippers and skin was cut with scissors. The collected skins were rapidly frozen with liquid nitrogen and grinded into powder with sandpaper. Total RNA was extracted from ISOGEN II (NIPPON GENE Co., Ltd., Toyama, Japan). The preparation of a cDNA library and sequencing was conducted at Rhelixa Co., Ltd. (Tokyo, Japan) with Illumina Nova Seq 6000. (Illumina, San Diego, CA, USA) RNA-Seq reads from these nine libraries were adapter-trimmed using Trim Galore! (ver. 0.6.10) [29] and cutadapt (ver. 4.4) [30]. The cleaned reads were subsequently mapped to the *R. norvegicus* genome assembly (Rnor_6.0 [31]) using HISAT2 (ver. 2.1.0) [32] with default parameters. Gene expression quantification was performed using StringTie (ver. 2.2.1) [33] and prepDE.py (http://ccb.jhu.edu/software/stringtie/dl/prepDE.py accessed on 1 October 2024) with a GTF file (Rattus_norvegicus. Rnor_6.0.104 [31]) that was used to generate the count matrix. The count data were normalized by the trimmed mean of M values (TMM) method provided in the TCC library (ver. 1.42.0) [34,35]. To cluster genes based on similar expression patterns, hierarchical clustering analysis was conducted using the edgeR library (ver. 4.0.16) based on the normalized count data [36]. To identify differentially expressed genes (DEGs) between samples (WT vs. KO, WT vs. KI, and KO vs. KI), we used the TCC package with default options, in which the TMM method and edgeR-based DEG analysis were employed. A Venn diagram was generated to illustrate the significant DEGs (FDR < 0.05) shared between WT vs. KO and WT vs. KI comparisons, utilizing the VennDiagram package (ver. 1.7.3) [37]. Additionally, the normalized count data were used to compare the gene expression levels between WT, KI, and KO rats, which were visualized in dot plots.

### 4.7. Statistical Analysis

The statistical significance of differences in Figure 8 and Figure 12 was analyzed by the Student’s *t*-test. *: *p* < 0.05, **: *p* < 0.01, ***: *p* < 0.001, n.s.: not significant (Appendix A). The statistical significance of differences in Figure 6, Figure 9, Figure 10 and Figure 11 was analyzed by the Tukey-Kramer procedure with one-way ANOVA. *: *p* < 0.05, **: *p* < 0.01, ***: *p* < 0.001, n.s.: not significant (Appendix A).

## 5. Conclusions

Comparison of WT, VDR-KO, and our novel KI rats, which have VDR completely lacking ligand-binding ability, clearly demonstrated that non-liganded VDR plays a critical role in the differentiation, proliferation, and apoptosis of keratinocytes in both hair follicles and the epidermis.

## Figures and Tables

**Figure 1 ijms-26-00422-f001:**
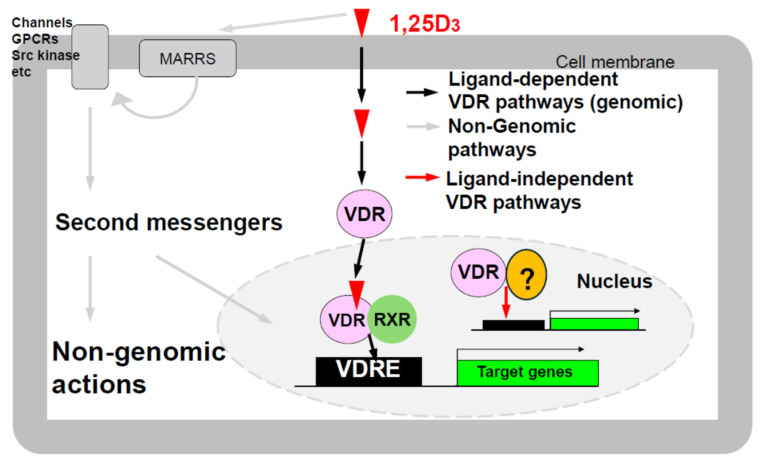
Multiple mechanisms of active vitamin D and VDR actions. MARRS: An abbreviation for membrane-associated rapid response sterol binding protein. The question mark indicates a protein factor that forms complexes with non-liganded VDR, with Hairless (Hr) being one known example.

**Figure 2 ijms-26-00422-f002:**
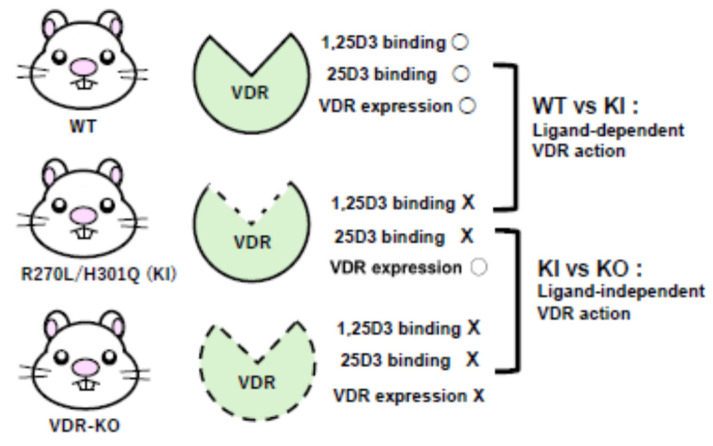
The three types of rats used in this study. WT: wild-type rats, KI: rats expressing mutant *Vdr*-R270L/H301Q), KO: *Vdr* gene-deficient rats. The 1,25D3 and 25D3 cannot bind to *Vdr*-R270L/H301Q. The expression level of *Vdr*-R270L/H301Q in KI rats is nearly the same as that of normal *Vdr* in WT rats. By comparing WT and KI rats, ligand-dependent actions of VDR can be highlighted, and by comparing KI and KO, the functions of non-liganded VDR can be delineated.

**Figure 3 ijms-26-00422-f003:**
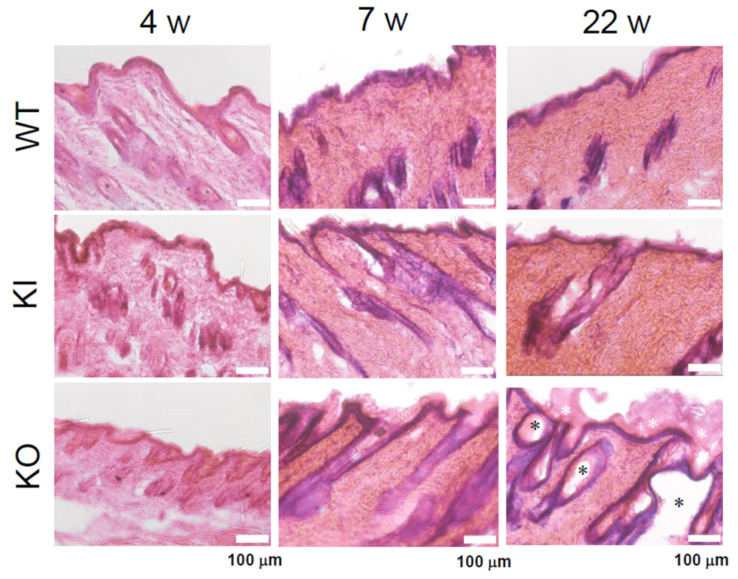
H&E-staining of dorsal skin of KI and KO rats at 4, 7, and 22 weeks of age.

**Figure 4 ijms-26-00422-f004:**
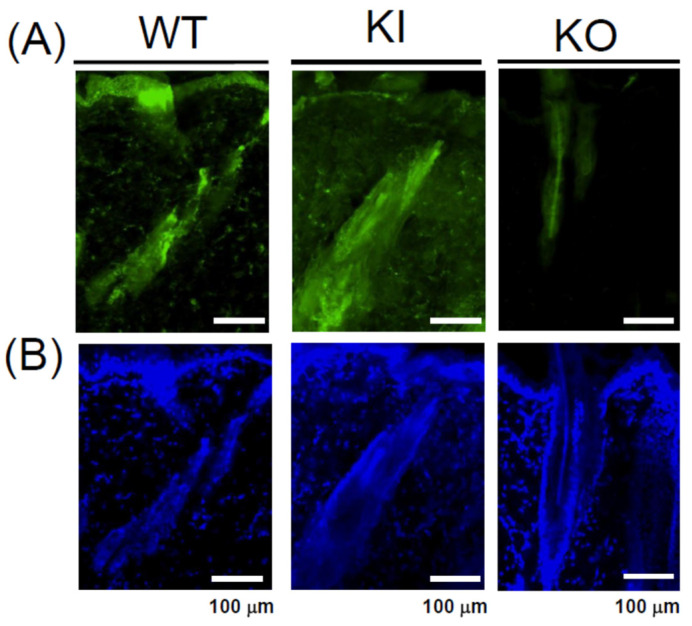
Immunostaining analysis of VDR (**A**) and DAPI staining (**B**) in dorsal skin of WT, KI, and KO rats.

**Figure 5 ijms-26-00422-f005:**
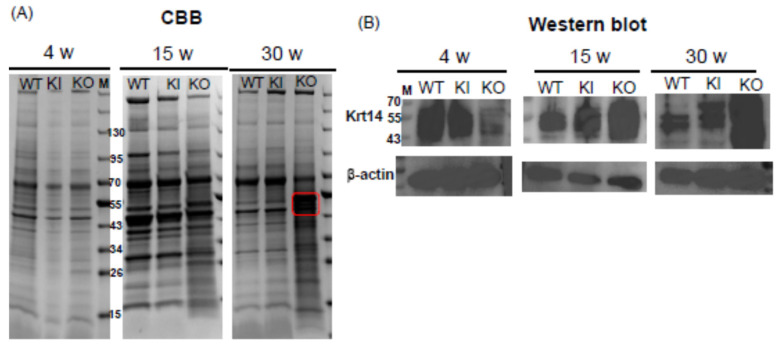
SDS-PAGE (**A**) and Western blot analysis of Krt14 (**B**) in dorsal skin protein prepared from WT, KI, and KO rats at 4, 15, and 30 weeks of age. M: Thermo Scientific PageRuler Prestained NIR Protein Ladder (Waltham, MA, USA). The red area indicates where an extremely intense bands were observed exclusively in KO.

**Figure 6 ijms-26-00422-f006:**
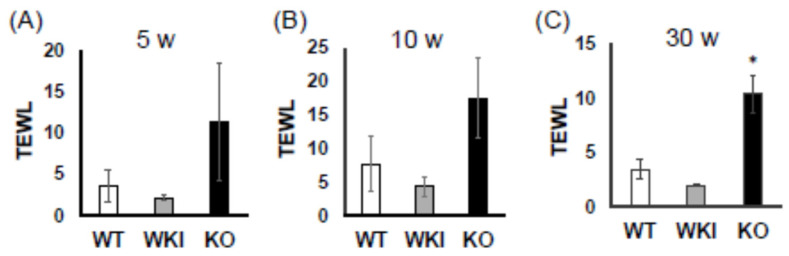
Comparison of transepidermal water loss in the skin of WT, KI, and KO rats at 5 (**A**), 10 (**B**), and 30 (**C**) weeks of age. A significant difference was observed between WT and KO in (**C**). The statistical significance of differences was analyzed by the Tukey-Kramer. *: *p* < 0.05.

**Figure 7 ijms-26-00422-f007:**
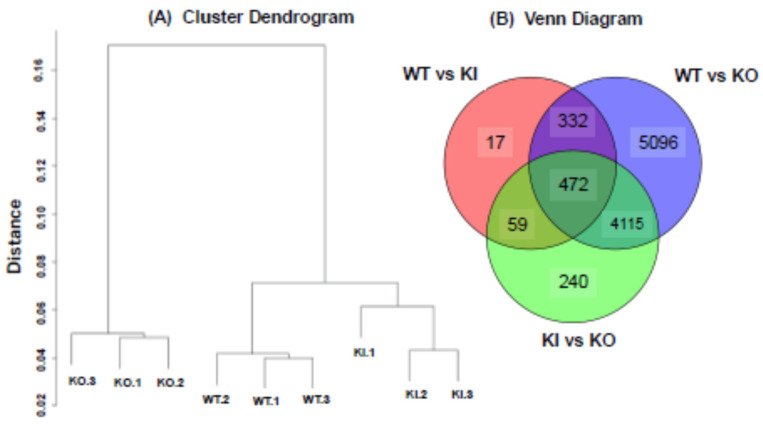
Cluster diagram of WT, KI, and KO rats (**A**) and Venn diagram among WT vs. KO, WT vs. KI, and KI vs. KO (**B**).

**Figure 8 ijms-26-00422-f008:**
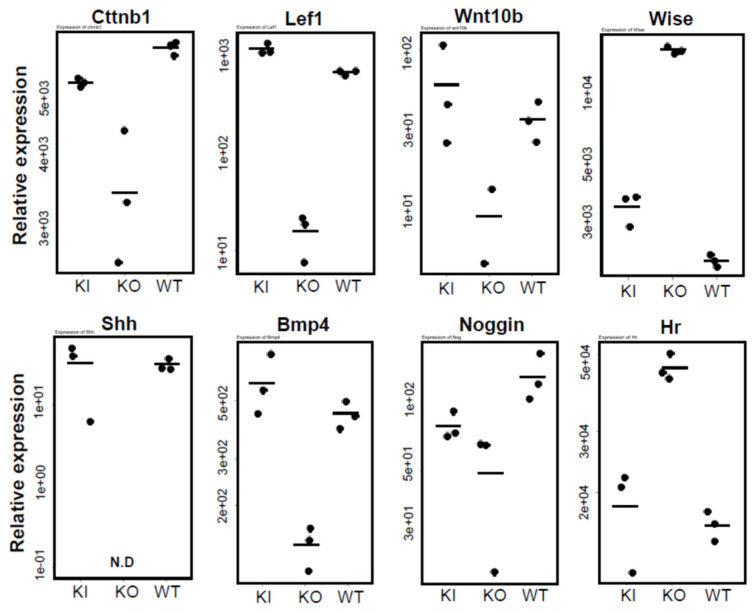
Comparison of mRNA levels of genes required for hair follicle development and regeneration among WT, KO, and KI rats. The numbers on the vertical axis, such as 5e+03, represent 5 × 10^3^. The statistical significance of differences was analyzed by the Student’s *t*-test (see Appendix A).

**Figure 9 ijms-26-00422-f009:**
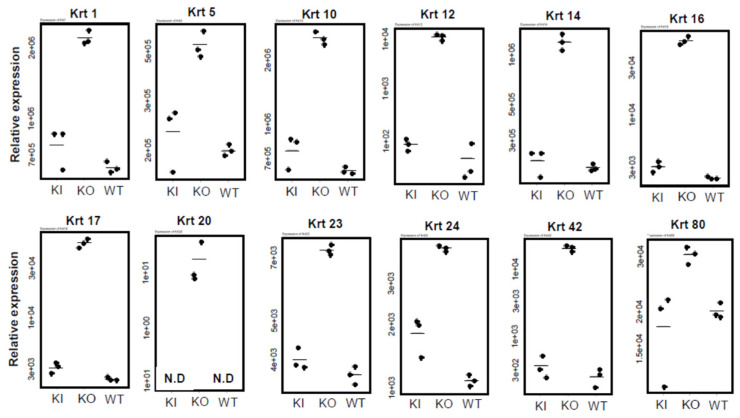
Comparison of mRNA levels of epidermal keratins among WT, KO, and KI rats. The numbers on the vertical axis, such as 5e+03, represent 5 × 10^3^. The statistical significance of differences was analyzed by Tukey-Kramer procedure with one-way ANOVA (see Appendix A).

**Figure 10 ijms-26-00422-f010:**
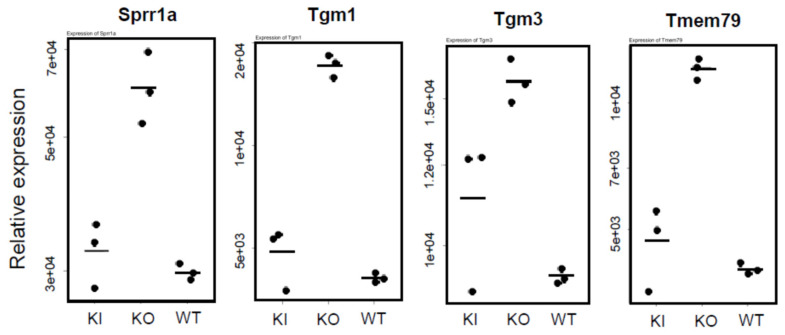
Comparison of mRNA levels of keratinization-related genes among WT, KO, and KI rats. The numbers on the vertical axis, such as 5e+03, represent 5 × 10^3^. The statistical significance of differences was analyzed by the Tukey-Kramer procedure with one-way ANOVA (see Appendix A).

**Figure 11 ijms-26-00422-f011:**
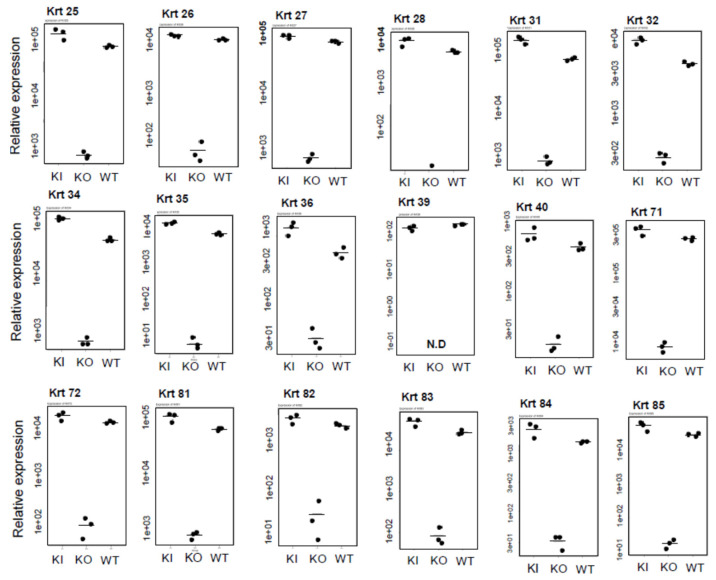
Comparison of mRNA levels of hair keratins and inner root sheath keratins among WT, KO, and KI rats. The numbers on the vertical axis, such as 5e+03, represent 5 × 10^3^. The statistical significance of differences was analyzed by Tukey-Kramer procedure with one-way ANOVA (see Appendix A).

**Figure 12 ijms-26-00422-f012:**
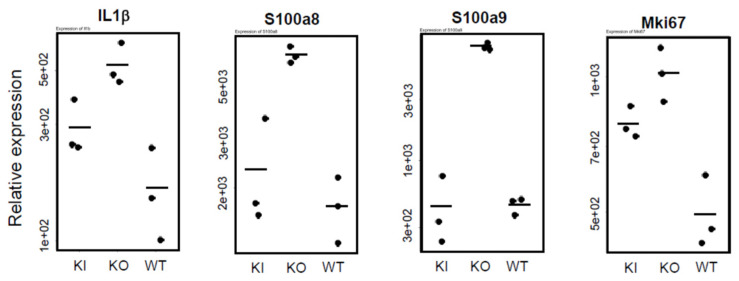
Comparison of mRNA levels of genes involved in inflammation (*IL1β*, *S100a8*, *S100a9*) or cell proliferation (*Mki67*) among WT, KO, and KI rats. The numbers on the vertical axis, such as 5e+03, represent 5 × 10^3^. The statistical significance of differences was analyzed by the Student’s *t*-test (see Appendix A).

**Table 1 ijms-26-00422-t001:** Antibodies used in Western blot analysis.

Protein	First Ig	Second Ig
VDR	Anti-VDR antibody (D2K6W) Rabbit mAb (Cell Signaling Technology)	Alexa Flour 488 goat anti-rabbit IgG (Invitrogen)
Krt14	Anti-Cytokeratin 14 polyclonal rabbit Ig (Proteintech Group, Inc., Rosemont, IL, USA)	HRP-conjugated goat anti-rabbit #7074 (Cell Signaling Technology)
Krt1, 5, and 8	clone PCK-26, purified from hybridoma cell culture C5992 (Sigma-Aldrich, St. Louis, MO, USA)	HRP-conjugated horse anti-mouse IgG(H+L) #7076 (Cell Signaling Technology)
β-actin	β-Actin Antibody #4967 (Cell Signaling Technology)	HRP-conjugated Anti-rabbit #7074 (Cell Signaling Technology)

## Data Availability

The datasets generated or analyzed during the current study, including raw bulk data of RNAseq are available from the corresponding author (T.S.) on reasonable request. A genomic sequence of the *Vdr* gene of *Vdr*-KO rats containing the mutated position is available in the DDBJ database accession number 370 LC764592 (http://getentry.ddbj.nig.ac.jp/top-j.html accessed on 1 October 2024).

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
