# Peer review of "Ligand-Independent Vitamin D Receptor Actions Essential for Keratinocyte Homeostasis in the Skin"

_ijms, 2025, doi:10.3390/ijms26010422_

Round 1
Reviewer 1 Report
Comments and Suggestions for Authors
Comments to the Authors
The manuscript investigates the role of ligand-independent vitamin D receptor (VDR) actions in keratinocyte homeostasis using genetically modified rat models. The use of VDR-mutant (R270L/H301Q) rats is an important tool, enabling for a clear differentiation between ligand-dependent and ligand-independent actions. The combination of RNA-Seq, protein analysis, and histological techniques provides comprehensive data to support the conclusions.
The study is innovative and contributes to the understanding of VDR's non-canonical roles in skin physiology and pathology, particularly focusing on alopecia and skin barrier integrity. However, certain aspects require clarification, restructuring, or further discussion to strengthen the manuscript. Addressing the concerns outlined below will significantly enhance the manuscript impact.
Major Concerns
1. All experiments were conducted exclusively on male rats. Considering potential sex differences in VDR-mediated actions, the inclusion of female rats would strengthen the results.
2. Figure 3-4: The images are of suboptimal resolution and with poor color contrast, making VDR localization challenging to visualize. Improving image quality and providing scale bars in Figure 3 is recommended for accurate evaluation. Provide detailed legends is necessary to improve clarity. Please include also the DAPI (or nuclear staining) to better visualize the VDR intracellular localization. A quantitative analysis of hyperkeratosis and cyst size would add rigor to the observations.
3. Figure 5: Protein bands appear faint, making comparisons across groups challenging. Including loading controls for normalization in all blots is essential.
4. While transepidermal water loss (TEWL) (Figure 6) was measured, additional functional assays would strengthen the data. For example, histological correlation with barrier impairment and/or electron microscopy to visualize ultrastructural changes could provide additional insights into the biophysical properties of the skin barrier.
5. The RNA-seq data reveal differential gene expression, but further validation (e.g., qRT-PCR or immunohistochemistry) for key genes in the Wnt and BMP pathways is necessary to substantiate claims. In addition, while the overlap of DEGs is presented, it lacks on the functional categories or pathways enriched in these gene sets. Adding GO or KEGG pathway enrichment analyses would provide context to the findings. The plots (Figures 8, 9, 10, 11) are informative but lack statistical details. Adding statistical tests directly in figure legends would enhance interpretability (please increase the quality of the plot).
6. The role of inflammation-related genes (e.g., IL-1β) is mentioned but underexplored. Further experiments such as cytokine profiling or immunohistochemistry for immune cell infiltration would confirm inflammatory changes in KO rats.
7. The statistical tests used are appropriate but not consistently described in figure captions or the main text. For example, Figures 8 and 12 rely on p-values but lack annotation of specific comparisons (e.g., WT vs. KO, WT vs. KI).
Minor comments:
1. Typos and grammatical inconsistencies appear throughout (e.g., "sizzlers" instead of "scissors" in methods).
2. Expand the discussion to include broader implications of findings while integrating recent literature.While the data support some mechanistic insights, certain claims (e.g., cross-talk between pathways) lack direct experimental validation. The discussion does not adequately address the translational implications of these findings for skin-related disorders or alopecia in humans.
Reviewer 2 Report
Comments and Suggestions for Authors
This manuscript explored the mechanisms of VDR in maintaining skin homeostasis. Kindly address the commons below will enhance the quality of the manuscript.
1. Pathway color in figure 1 is hard to distinguish, please change color to make them easy to tell
2. Please include scale bar for the figure 3.
3. Western blot in Figure 5 and supplemental Fig.2 are over saturated, please use better pictures.
4. Figure 5 (30w) and supplemental figure S2 share the same actin blot?
5. Figure 8, figure 9, figure 10, figure 11 and figure 12 are very blurry to read, please replace with higher quality picture.
6. Why this question is important to address? What are the implications and benefits of studying this topic?
7. What is most significantly changed pathway between KI and KO? Any important clue for VDR’s Ligand independent role when comparing between?
8. What is the most significantly changed protein between KI and KO?
9. What transcriptional factor is most changed between KI and KO?
10. Please include the raw bulk RNA -seq data as supplement.

The English could be improved to more clearly express the research.
